# The Relationship between Soil Particle Size Fractions, Associated Carbon Distribution and Physicochemical Properties of Historical Land-Use Types in Newly Formed Reservoir Buffer Strips

**Tengfei Yan [1,2,3], Yevheniia Kremenetska [1,*], Biyang Zhang [3], Songlin He [2,*], Xinfa Wang [1,2] , Zelong Yu [3], Qiang Hu [3], Xiangpeng Liang [3], Manyi Fu [3] and Zhen Wang [3]**

[1] Faculty of Agrotechnologies and Natural Resource Management, Sumy National Agrarian University, 40000 Sumy, Ukraine; 2011290003@xyafu.edu.cn (T.Y.); wangxf@hist.edu.cn (X.W.)
[2] School of Horticulture Landscape Architecture, Henan Institute of Science and Technology, Xinxiang 453003, China
[3] Department of Forestry, Xinyang Agriculture and Forestry University, Xinyang 464000, China; 2018290001@xyafu.edu.cn (B.Z.); yuzelong05@163.com (Z.Y.); 2018290002@xyafu.edu.cn (Q.H.); l1419693015@126.com (X.L.); fmy556@xyafu.edu.cn (M.F.); wz062618@163.com (Z.W.)
* Correspondence: e.kremenetska@gmail.com (Y.K.); hsl213@yeah.net (S.H.)

**Abstract:** Water impoundment reshapes the ecological environment around the bank-line of newly built reservoirs. Therefore, reservoir buffer strips play a disproportionately large role in the maintenance of ecosystem functions and environmental benefits during the early stage of reservoir formation. However, there are gaps in the research on soil particle-size-associated carbon distribution characteristics within different historical land-use types in newly formed reservoir buffer strips. In this study, we focused on soil particle size fractions, aggregate stability, and particle-size-associated carbon distribution characteristics of different historical land-use types of reservoir buffer strips at distance scale (i.e., different distance from the water) after reservoir impoundment in the Chushandian Reservoir, China, and explored the relationship between them. The results showed that the soil texture of abandoned cropland and grassland are classified as silt loam and woodland are classified as sandy loam; different historical land-use types in newly formed reservoir buffer strips showed significant differences in soil aggregate stability after reservoir impoundment; a distance scale was used to measure these differences, which were mainly due to the dry-wet cycles and water submerged condition caused by the buffers' different distances from water. The newly formed reservoir buffer strips underwent corresponding changes in the particle-size-associated carbon distribution characteristics after reservoir impoundment, mainly due to the turnover property of different soil particles combined with organic carbon. Reservoir impoundment accelerates the turnover of silt particle and associated nutrients in soils of historical land-use types in newly formed reservoir buffer strips; turnover may be mediated mainly by microbial biomass.

**Keywords:** dam construction; silt particle; Chushandian reservoir; turnover property; carbon storage capacity

## 1. Introduction

Dam construction significantly alters the ecological environment around rivers [1]; newly formed reservoir habitats due to water impoundment have far-reaching effects on vegetation development [2–4], nutrient cycling [5,6], soil aggregates stability [7,8], and microbial communities [9,10] in the surrounding riparian zone. Therefore, reservoir buffer strips between the baseline (normal) level and terrestrial systems are important in maintaining ecosystem functions and environmental benefits during the early stages of reservoir formation [11].

Reservoir buffer strips are an essential part of the reservoir buffer zone, which is often narrow but is critical in detaining non-point pollution, purifying water quality, holding soil and water, and carbon sequestration [12–15].

The widened reservoir bank often occupies a large area of agriculture land and woodland, and the reservoir environment after impoundment reshapes habitats in the surrounding historical land-use types, prompting the gradual transition from terrestrial to aquatic ecosystems [16,17]. In addition, the shaping of the bank environment by reservoir impoundment is a dynamic process, so it is important to monitor the early habitat response of the reservoir buffer strips for the restoration of the ecological environment surrounding reservoirs. Previous studies have paid attention to the configuration pattern and nutrient retention capacity of reservoir buffer strips [18–21], but there are gaps in the distribution characteristic of soil organic carbon in soil particles of historical land-use types after reservoir impoundment.

The capacity of soils to store organic carbon, an essential functional factor of ecosystems, is sensitive to disturbances and land management, and the distribution of soil particles (especially silt and clay content) and aggregate stability are critical evaluation indicators [22,23]. The stability mechanism of soil organic carbon (SOM) by soil particles is mainly achieved because of mineral surfaces, and the contribution of different particle size fractions to soil organic matter varies [24]. For example, clay particle combined with SOM can effectively isolate microbial decomposition and form a stable carbon pool. Silt particle, in contrast, is frequently considered to be a carbon pool for rapid turnover of SOM [25,26]. Therefore, the distribution characteristics of organic matter within soil particles are regularly used to evaluate the ability of soil to resist external stress and carbon sequestration [27,28].

Dramatic changes in soil moisture conditions are the main driving force of soil geochemical processes that affect different historical land-use types in reservoir buffer strips [29]. After reservoir impounding, water levels in the buffer strips around the reservoir bank noticeably rise, soil redox status significantly changes, dry-wet soil conditions undergo a complex cyclic process, and the capacity of soil to store organic carbon and the stability of aggregates change accordingly [30–32]. Also, the response of soil carbon storage capacity to water impoundment of different historical land-use types in reservoir buffer strips varies depending on distance from water because of the slope length and dam-triggered flooding intensity [33]. Frequent dry-wet alternations cause fragmentation of soil aggregates, promote mineralization of organic carbon and nitrogen, and accelerate the release of organic matter [8,30]. All indications show that without human intervention, the landscape of different historical land-use types in reservoir buffer strips will gradually degrade after reservoir impoundment, which greatly threatens the stability of the reservoir bank ecosystem.

The relationship between soil properties, SOM decomposition and stabilization, and soil aggregates dynamics in different terrestrial ecosystems has been intensively studied [24,34]; however, in reservoir buffer strip ecosystems, especially in newly built reservoirs, this relationship has yet to be deepened [33]. Although many reports have shown that metal sesquioxides, such as the oxyhydroxides of Al and Fe and exchangeable cation concentrations ($Ca^{2+}$, $Mg^{2+}$, etc.), are closely related to aggregate stability and organic matter distribution [35–37], the relationship between soil physicochemical properties and soil particle-size fractions and associated nutrient distribution is not fully recognized. This lack of understanding of the relationship is mainly due to the fact that, on one hand, the evaluation indicators of soil functional traits are complex and diverse, and the inclusion of different indicators may yield completely opposite results [22,38], and on the other hand, change in soil properties are mainly driven by external factors. Numerous studies have shown that natural dry-wet cycles affect the relationship between aggregates, granular organic matter, and riparian microbial communities [39]. Similarly, soil nutrient cycles, such as those for carbon [40], nitrogen, and phosphorus [41], can change as well [31]. Therefore, the incorporation of additional soil physicochemical property indicators is relevant

for a more comprehensive discussion of the role of soil properties in the stability of soil aggregates [42].

Located in Xinyang City, Henan Province, China, the Chushandian Reservoir is a large reservoir with flood control and irrigation functions in the Huai River catchment. The stored water reached its baseline (normal) level for the first time in October 2020. During the construction of the reservoir, a large area of cropland and commercial woodland need to be occupied due to the widening of the reservoir bank, and nearby households adopted different countermeasures for land management. After reservoir impoundment, a mosaic landscape dominated by abandoned cropland, grassland, and commercial woodland formed in the reservoir buffer strips. Therefore, it is important to study the soil particle-size fractions and associated carbon distribution characteristics of different historical land-use types after reservoir impoundment to monitor soil quality and ecological evolution of the reservoir buffer strips and to provide a scientific basis for next-step restoration measures.

In this study, we refer to the criteria of the soil texture classification system to group soil particles [14]. Soils in reservoir buffer strips are sticky and heavy due to the effect of water level uplift (the local soil texture type was mainly ACRISOLS soil, based on FAO World Reference Base), and consequently, effective mechanical separation is challenging because the air-dried soil is almost glued into a complete macro-aggregate. Hence, using soil particle size fractions to characterize soil aggregates is a more effective method. To the best of our knowledge, no study has attempted to characterize the particle size fractions and associated organic carbon distribution in soils of historical land-use types in buffer strips of a newly built reservoir, especially in the context of reservoir impoundment, how do the distribution characteristic of different soil particle-sizes associated carbon play a role in the turnover of soil carbon pool, and what is the relationship between the distribution characteristics of soil particle-size-associated carbon and physicochemical properties? Thus, the main objective of this study is to document the particle-size fractions, aggregates' stability, and particle-size-associated carbon distribution characteristics of different historical land-use types in newly formed reservoir buffer strips after water impoundment and to explore the relationship between soil particle size fractions and associated carbon distribution and physicochemical properties. We hypothesize that the response of soil aggregates stability to water impoundment differs on the distance scales in the newly formed reservoir buffer strips, and that this difference is mainly caused by a dry-wet cycle at different distances from water; in addition, we assume that water impoundment will change the binding characteristic of soil particles with organic carbon in newly formed reservoir buffer strips, and there is a close relationship with soil physicochemical properties.

## 2. Materials and Methods

### 2.1. Field Site

We conducted our field study in the riparian zone of the Chushandian Reservoir (N32°22′–32°32′, E113°89′–113°96′) located upstream on the Huai River in Xinyang city in Henan province, China. The climate of the Chushandian Reservoir is a transitional region from a subtropical zone to a warm temperate zone. Our previous article [43] describes the specific climatic and vegetation characteristics of the Chushandian Reservoir.

### 2.2. Sample Collection

We collected soil samples from 11 December 2020, two months after the water level first reached its normal level, and we chose three historical land-use types (abandoned cropland, grassland, and woodland) on the west side of the reservoir, which had no dyke protection. Three sample plots were selected for each land-use type (9 in total), and the distance between the sample plots did not exceed 1.5 km (Figure 1). The cropland was owned by the residents before the dam was built, but it was abandoned due to the rising water level after the dam was constructed. Up to the time of sampling, the abandoned cropland had naturally recovered over several months. The main vegetation was *Veronica didyma* Tenore, *Conyza canadensis* (L.) Cronq., and *Alternanthera philoxeroides* (Mart.) Griseb. The grassland

was previously abandoned cropland (no agricultural production during the initial stage of reservoir construction due to the relocation of residents). At the time of sampling, the sample site had undergone 4–5 years of natural recovery, and the main vegetation was *Imperata cylindrica* (L.) Beauv., *Xanthium sibiricum* Patrin ex Widder, and *Conyza canadensis* (L.) Cronq. In the woodland, the main species of tree was chestnut (*Castanea mollissima* BL.), the understory was the tea variety Xinyang Maojian (*Camellia sinensis* (L.) O. Ktze.), and the main vegetation was *Imperata cylindrica* (L.) Beauv. and *Carex* Linn., *Phyllostachys glauca* McClure. The three historical land-use types characterize different successional time series. The primitive characteristics of the soil in three land-use types before reservoir construction were pH: 7.1; TC (total carbon): 8.32 g/Kg; EC (electric conductivity): 55.40 $\mu S \cdot cm^{-1}$, respectively [44]. The sample strip distance was set according to the plot size of different historical land-use types along the reservoir bank. As the different historical land-use types were previously operated by patchy farmers, the landscape of the reservoir buffer strips after water impoundment was fragmented, and the width of independent land types was mostly about 25–30 m. To eliminate the influence of boundary effects of adjacent sample plots, we set up sample strips of 0 m, 2 m, 20 m within different historical land-use types along the bank. We labeled each sample strip according to different historical land-use types as abandoned cropland (C0, C2, C20), grassland (G0, G2, G20), and woodland (W0, W2, W20); we collected 27 samples in total (Figure 1). The 0 m strip was set at the junction of the water and land, where it remained submerged in water most of the time, and here sampling was carried out using a Peterson mud collector. For the other sample plots, we used a soil sampler to collect 5 soil samples from a depth of 10 cm and combined them to form one soil sample, which we then placed in a self-sealed bag to be transported to the laboratory immediately.

We divided each composite sample into two parts, keeping one part in a 4 °C refrigerator to determinate the microbial biomass carbon (MBC), microbial biomass nitrogen (MBN), and dissolved organic carbon (DOC). We placed the other part in a ventilated area to dry naturally and gently broke it into aggregates along natural cracks and used forceps to remove roots, stones, and large macrofauna during the drying period. Since the soil was relatively sticky and heavy, it proved difficult to break into natural aggregates completely, so we used a wooden stick to crush the soil slightly, trying not to destroy the particle size fractions. After, we passed all crushed soil samples through a 2 mm sieve and further separated the result into two parts. One was passed through a 0.25 mm sieve to determine soil total carbon (STC), soil total nitrogen (STN), and soil total phosphorus (STP), and the other one was used to determine the particle size fractions.

### 2.3. Physicochemical Properties of Soils

We used an elemental analyzer (Vario MAX C/N, Elementar Analysensysteme GmbH, Hanau, Germany) to determine the level of STC and STN in our soil samples, and soil pH was measured in distilled water mixed 2.5:1 (by volume) with dry soil using a Delta 320 pH meter (Mettler-Toledo Instruments (Shanghai) Ltd., Shanghai, China). We determined soil STP content by using the molybdenum-blue colorimetry method after digesting the samples in perchloric acid. We extracted DOC with $K_2SO_4$ and determined its level by dichromate digestion. MBC and MBN were measured with the fumigation extraction method [45,46], and we measured bulk density (BD) by using the oven-drying volumetric ring method after samples were oven-dried at 105 °C for 24 h to a constant mass. We then calculated BD as the ratio of the oven-dried undisturbed core weight to the cutting ring volume. Soil moisture (SM) content was determined by oven-drying the samples at 105 °C for 24 h, and water content was expressed as a percentage of the dry weight. The physicochemical properties of different soils are shown in Table 1.



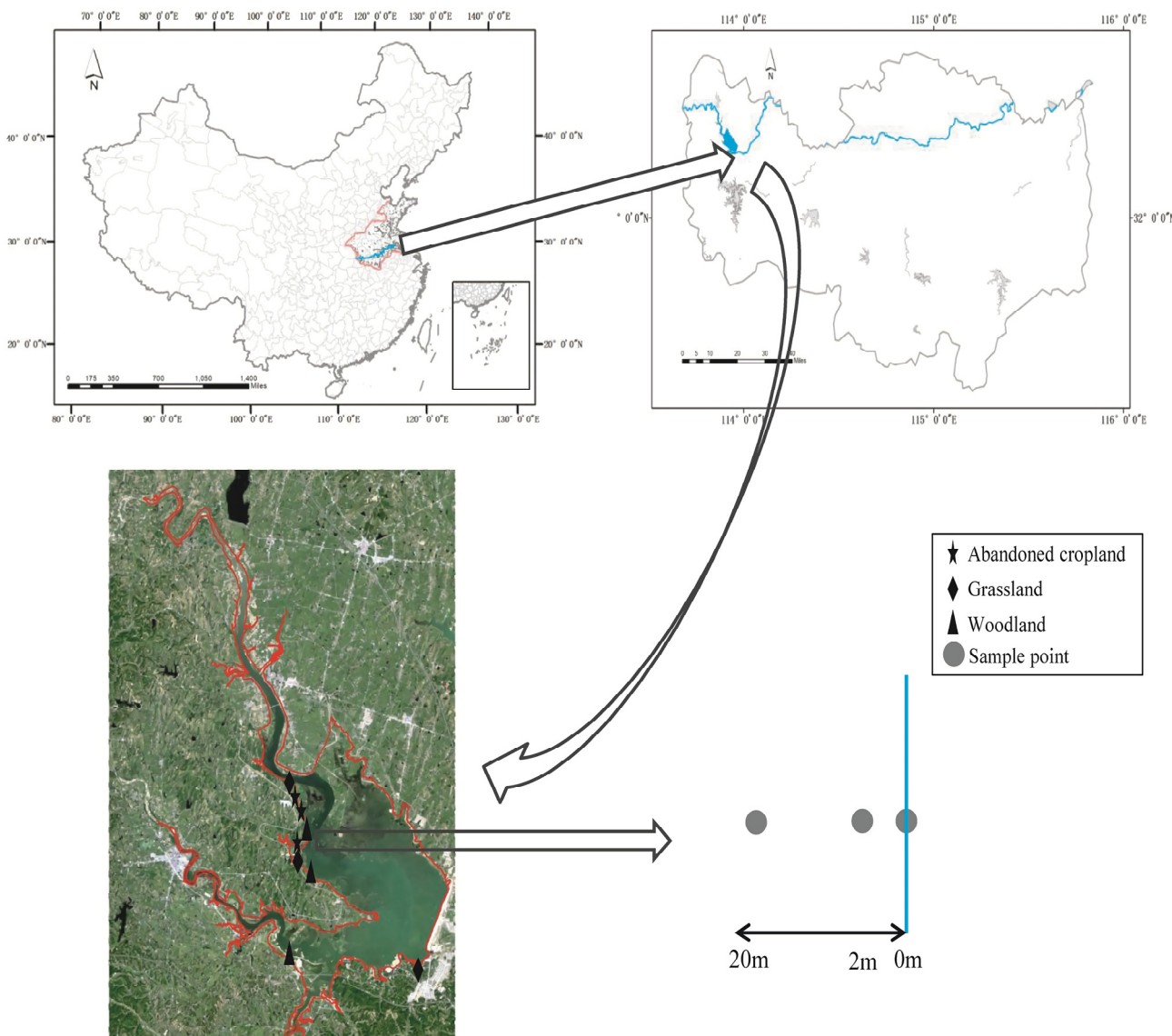

**Figure 1.** Location of study area: (**top left**) boundary map of China; (**top right**) boundary map of Henan province; (**bottom left**) locations of the sampling sites in the riparian zone of the Chushandian Reservoir, China; (**bottom right**) sample strip setting diagram. The pentagram represents the abandoned cropland; the diamond represents the grassland; and the triangle represents the woodland. The circle represents the setting of sampling strips.

**Table 1.** Physicochemical properties of soil from different historical land-use types in newly formed reservoir buffer strips after water impoundment. Different lowercase letters indicate significance between different soil sampling sites ($p < 0.05$).

| | Cropland | | | Grassland | | | Woodland | | |
|---|---|---|---|---|---|---|---|---|---|
| | C0 | C2 | C20 | G0 | G2 | G20 | W0 | W2 | W20 |
| pH | 6.35 ± 0.19 a | 5.92 ± 0.18 ab | 5.48 ± 0.37 ab | 5.83 ± 0.07 ab | 5.18 ± 0.10 b | 5.28 ± 0.09 ab | 6.15 ± 0.10 a | 5.86 ± 0.08 ab | 6.15 ± 0.37 ab |
| BD | NA | 1.41 ± 0.12 a | 1.32 ± 0.02 a | NA | 1.35 ± 0.04 a | 1.36 ± 0.08 a | NA | 1.18 ± 0.10 ab | 1.02 ± 0.04 b |
| STC | 9.04 ± 0.91 b | 10.18 ± 1.30 b | 8.46 ± 0.34 b | 8.12 ± 0.32 b | 7.25 ± 0.35 b | 5.63 ± 0.19 b | 8.65 ± 0.89 b | 29.08 ± 4.01 a | 9.60 ± 1.38 b |
| STN | 1.03 ± 0.13 b | 1.02 ± 0.08 b | 0.90 ± 0.05 b | 0.92 ± 0.02 b | 0.80 ± 0.05 b | 0.64 ± 0.03 b | 0.89 ± 0.04 b | 2.13 ± 0.28 a | 0.91 ± 0.09 b |
| STP | 0.27 ± 0.09 a | 0.32 ± 0.07 a | 0.42 ± 0.02 a | 0.42 ± 0.00 a | 0.38 ± 0.023 a | 0.23 ± 0.08 a | 0.26 ± 0.01 a | 0.28 ± 0.01 a | 0.45 ± 0.20 a |
| C/N | 8.84 ± 0.27 b | 9.88 ± 0.70 ab | 9.46 ± 0.39 b | 8.85 ± 0.21 b | 9.05 ± 0.38 b | 8.81 ± 0.12 b | 9.72 ± 0.54 ab | 12.18 ± 0.30 a | 10.57 ± 1.12 ab |
| DOC | 17.23 ± 1.28 b | 17.16 ± 0.73 b | 16.93 ± 1.73 b | 14.58 ± 0.81 b | 15.12 ± 0.17 b | 11.84 ± 0.69 b | 13.95 ± 0.04 b | 17.79 ± 1.06 ab | 23.38 ± 2.08 a |
| MBC | 180.29 ± 9.64 b | 145.04 ± 13.11 b | 186.10 ± 13.12 b | 146.19 ± 2.52 b | 137.82 ± 7.49 b | 148.58 ± 6.40 b | 146.80 ± 1.30 b | 152.88 ± 13.82 b | 418.89 ± 17.29 a |
| MBN | 18.75 ± 1.78 b | 19.85 ± 1.90 b | 19.36 ± 0.56 b | 23.67 ± 3.83 b | 17.63 ± 0.39 b | 21.79 ± 2.20 b | 20.42 ± 3.33 b | 24.83 ± 0.36 b | 35.99 ± 1.07 a |

### 2.4. Particle-Size Fraction and Associated Carbon Separation

In this study, we physically grouped soil particles by size with reference to the criteria of the soil texture classification system: >53 μm (sand particle), 2–53 μm (silt particle), and <2 μm (clay particle). We isolated particles from composite samples by using the method proposed by Tiessen (1982) [47] and Wu et al., (2004) [48]. First, we weighed 20 g of soil that passed through a 2 mm sieve into a 250 mL beaker, and then we placed the soil in a sonifier to discretize it fully for 30 min. Next, we poured the suspension into a 53 μm sieve and washed it with about 450 mL of distilled water until the washout became clear. The material left on the top of the sieve was sand particle (>53 μm) and some plant residue. After this we placed approximately 300 to 350 mL of the eluate into a 450 mL centrifuge tube and spun it at 760 rpm for 4 min using a Mandal RC5C centrifuge. We then poured the suspension into a collector and added 100 mL of distilled water. We then shook it and further centrifuged it at 550 rpm for 2 min, decanted the suspension, and combined it with the previous suspension. The latter process was performed at least 4 or more times until complete separation of the clay particle (<2 μm) from the silt particle (2 to 53 μm) is achieved. We placed all collected particles in an oven at 50 °C for 72 h and then weighed them. Finally, we determined the carbon and nitrogen content of each particle using an elemental analyzer (Vario MAX C/N, Elementar Analysensysteme GmbH, Hanau, Germany).

### 2.5. Soil Stability Calculation and Statistical Analysis

Mean weight diameter (*MWD*, μm) is used to evaluate soil aggregate stability, and we calculate *MWD* with the formula in Kemper and Rosenau (1986) [49]:

$$MWD = \sum_{i=1}^{3} Xi \times Mi$$

where $X_i$ is the mean diameter of the *i*th particle (μm), $M_i$ is the mass proportion of *i*th size fractions in aggregates (%), and we calculate organic C pool in soil particle size fractions (*OCP*, g/m$^2$) with the formula in Yang et al., (2007) [50]:

$$OCP = \sum_{i=1}^{3} (Ni \times OCi) \times B_d \times H \times 10$$

where $N_i$ is the mass proportion of the *i*th size fraction in whole soils (%), $OC_i$ is the organic C content in the *i*th size fraction (g·kg$^{-1}$), $B_d$ is the soil bulk density (g·cm$^{-3}$), and *H* is the soil thickness (cm).

All variables were tested for normality using the Shapiro-Wilk statistic, and those that did not conform to a normal distribution were transformed with the natural logarithm. We analyzed differences in soil particle size fractions, aggregate stability, particle-size associated carbon and nitrogen, and soil properties among different sampling sites using analysis of variance (ANOVA) with a least-significant test (LSD) post-hoc test ($p < 0.05$). Redundancy analysis (RDA) was conducted to explore the multivariate correlation between soil particle size fractions, MWD, and soil physicochemical properties, and we used soil physicochemical properties as the environmental variables. RDA was also conducted to explore the multivariate correlation between soil particles, their associated nutrients, and soil physicochemical properties, and we used soil physicochemical properties as the environmental variables. Stepwise multiple regression analysis was used to examine the linear relationship between soil particle size fractions, soil physicochemical properties, and particle-size associated nutrients; the partial regression sums of squares was used to determine the significance of each retention variable. Structural equation models (SEM) between historical land-use types, distance from water, sand particle-size, silt particle-size, and clay particle-size associated nutrients, soil microbial activity, and soil chemical properties were constructed with a partial least squares path model (PLS-PM) (soil physical

properties were not included due to lack of BD at 0 m). All statistical analysis and plotting were executed in R (version 4.0.3).

## 3. Results

The predominant soil particle size fractions in the abandoned cropland and grassland is silt particle (42.08 to 75.43%), and that the predominant soil particle size fractions in woodland is sand particle (29.37 to 80.26%), followed by silt particle (7.38 to 62.86%) (Table 2). We classify the soil texture of abandoned cropland and grassland as silt loam and woodland as sandy loam. Our MWD calculations for different historical land-use types showed that woodland > abandoned cropland > grassland, with woodland occupying a significantly larger area than the other two historical land-use types, and our OCP calculations for different historical land-use types showed W20 > abandoned cropland > grassland > W2, and the OCP content of W20 was significantly greater than that at all other sites.

**Table 2.** Mean weight diameter (MWD), Organic C pool (OCP), and particle size fractions from different historical land-use types in newly formed reservoir buffer strips after water impoundment. Different capital letters indicate significance between different soil particle-size fractions at the same site ($p < 0.05$); different lowercase letters indicate significance of soil particles, MWD and OCP between different sites ($p < 0.05$).

|  | Sand (53–2000 μm) | Silt (2–53 μm) | Clay (<2 μm) | MWD (μm) | OCP (g/m²) |
|---|---|---|---|---|---|
| C0 | 21.39 ± 9.60 Babc | 61.24 ± 9.64 Aab | 17.36 ± 6.84 Ba | 236.53 ± 96.54 bc | - |
| C2 | 13.70 ± 3.79 Babc | 66.98 ± 4.67 Aa | 19.31 ± 8.23 Ba | 159.21 ± 40.00 c | 3226.31 ± 453.67 b |
| C20 | 23.07 ± 1.77 Babc | 58.65 ± 2.67 Aab | 18.28 ± 1.54 Ba | 253.01 ± 17.55 bc | 2577.61 ± 291.02 bc |
| G0 | 9.32 ± 2.10 Bbc | 68.46 ± 4.28 Aa | 22.22 ± 2.58 Ba | 114.58 ± 20.48 c | - |
| G2 | 7.57 ± 1.28 Cc | 68.68 ± 2.93 Aa | 23.75 ± 2.89 Ba | 96.74 ± 12.99 c | 2382.44 ± 283.78 bc |
| G20 | 7.70 ± 2.79 Cc | 73.70 ± 5.33 Aa | 18.60 ± 7.99 Ba | 99.44 ± 30.01 c | 2012.17 ± 155.91 bc |
| W0 | 55.00 ± 12.03 Aa | 26.54 ± 12.05 Ac | 18.46 ± 1.46 Ba | 572.00 ± 120.24 a | - |
| W2 | 46.84 ± 16.72 Aabc | 39.31 ± 15.56 Abc | 13.86 ± 5.90 Ba | 491.65 ± 167.60 ab | 1685.19 ± 149.89 c |
| W20 | 51.49 ± 7.49 Aab | 38.41 ± 6.65 Abc | 10.10 ± 0.97 Ba | 539.14 ± 75.11 a | 4468.07 ± 641.82 a |

Different historical land-use types have different amounts of carbon, and carbon content varied depending on soil particle size. We found that the carbon content in grassland and abandoned cropland was the lowest in silt particle, which was significantly lower than that in clay particle at the C20, G0, G2, and G20 sites. We observed that carbon content of each particle increased with the decrease of particle size in woodland. In addition, we found that the carbon content in clay particle at W2 and W20 was significantly higher than that of sand, and the highest carbon content in sand particle was found at C2 (24.26 g/kg) and the lowest at W2 (6.40 g/kg). We saw the highest carbon content in silt and clay particles at W20 (23.56 g/kg and 28.62 g/kg, respectively), which was significantly higher than at any other site (Figure 2).

The distribution characteristics of soil particle-size-associated nitrogen on different historical land-use types and sites show a similar trend to soil particle-size associated carbon (Figure 3). Several studies have shown that carbon and nitrogen in soil are highly coupled [51,52]; therefore, this study focuses on the distribution of soil particle-size-associated carbon.

Using RDA analysis, we explored the relationship between soil particle size fractions, MWD, and soil physicochemical properties of different historical land-use types and showed that the interpretation of soil physicochemical properties, in terms of soil particle size fractions and MWD, was mainly concentrated on the first axis of RDA, reaching 34.36%. We found that pH, STC, STN, DOC, MBC, and MBN were positively correlated with sand particle and MWD; soil physicochemical properties indicators, sand particle, and MWD were mainly concentrated in the area where the woodland samples were distributed

and that silt particle and clay particle were mainly concentrated in the areas where the abandoned cropland samples were distributed (Figure 4).

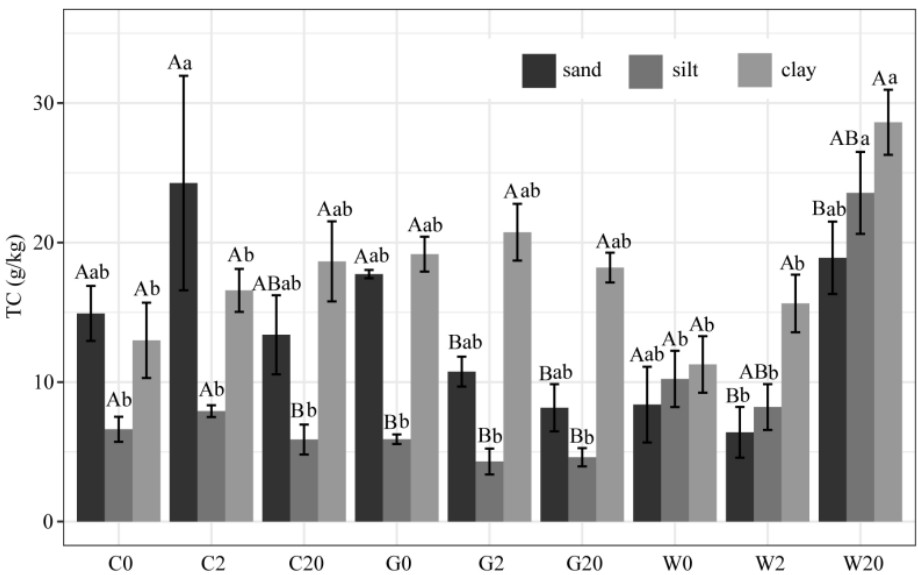

**Figure 2.** Characteristics of soil particle-size-associated carbon distribution under different historical land-use types in newly formed reservoir buffer strips after water impoundment. Different capital letters indicate significance between different soil particle size fractions at the same site ($p < 0.05$); Different lowercase letters indicate significance between different sites of the same particle size ($p < 0.05$). The vertical bars represent means, and the error bars represent standard errors ($n = 3$).

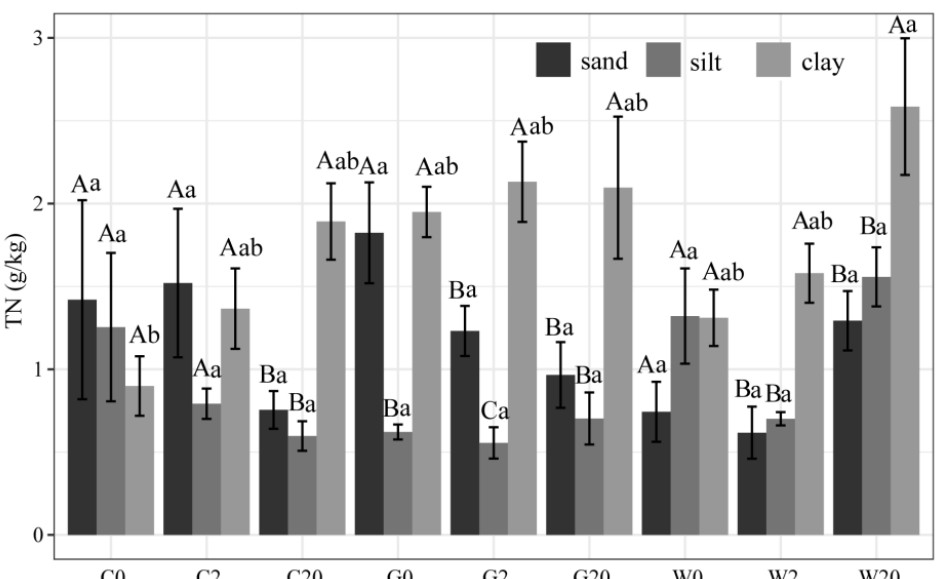

**Figure 3.** Characteristics of soil particle-size-associated nitrogen distribution in different historical land-use types in newly formed reservoir buffer strips after water impoundment. Different capital letters indicate significance between different soil particle-size fractions at the same site ($p < 0.05$); Different lowercase letters indicate significance between different sites of the same particle size ($p < 0.05$). The vertical bars represent means, and the error bars represent standard errors ($n = 3$).

Similarly, we also explored the relationship between soil particle-size-associated carbon, nitrogen distribution, and soil physicochemical properties in different historical land-use types by using RDA. We found that the degree of explanation is 19.1% on the first axis and 17.48% on the second axis of RDA. pH, STC, STN, DOC, and MBC were positively correlated with TC2 (silt particle-size-associated carbon) and TN2 (silt particle-size-associated

nitrogen) and were mainly concentrated in the areas where the woodland samples were distributed; TC3 (clay particle-size-associated carbon) and TN3 (clay particle-size-associated nitrogen) were concentrated in the areas where the abandoned cropland samples were distributed, and TC1 (sand particle-size-associated carbon) and TN1 (sand particle-size-associated nitrogen) were concentrated in the areas where the grassland samples were distributed (Figure 5).

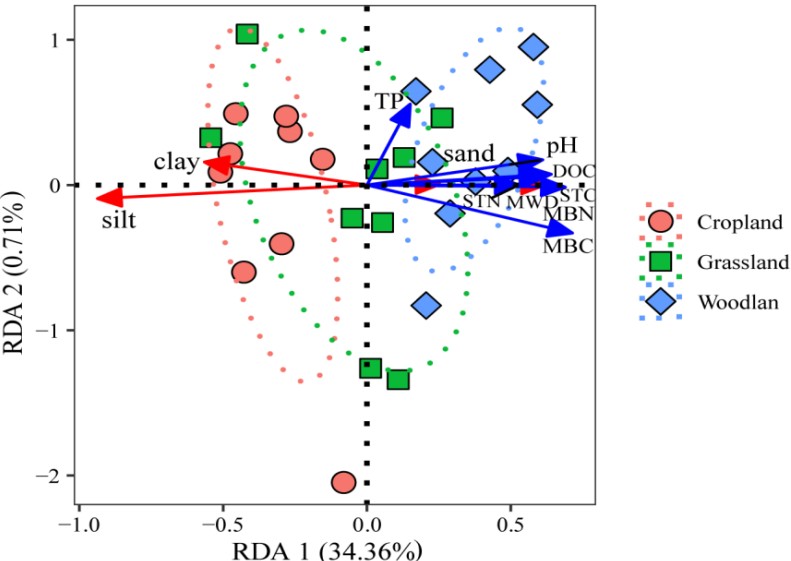

**Figure 4.** Ordination plot of the redundancy analysis (RDA) for the soil particle-size fractions and MWD with soil physicochemical properties as the explaining variables. The solid points indicate abandoned cropland scores; solid squares indicate grassland scores; solid rhombuses indicate woodland scores. The ellipses indicate the standard errors (*n* = 9) of different historical land-use types with a 95% confidence level. MWD is mean weight diameter; STC is soil total carbon; STN is soil total nitrogen; TP is soil total phosphorus; DOC is dissolved organic carbon; MBC is microbial biomass carbon; and MBN is microbial biomass nitrogen.

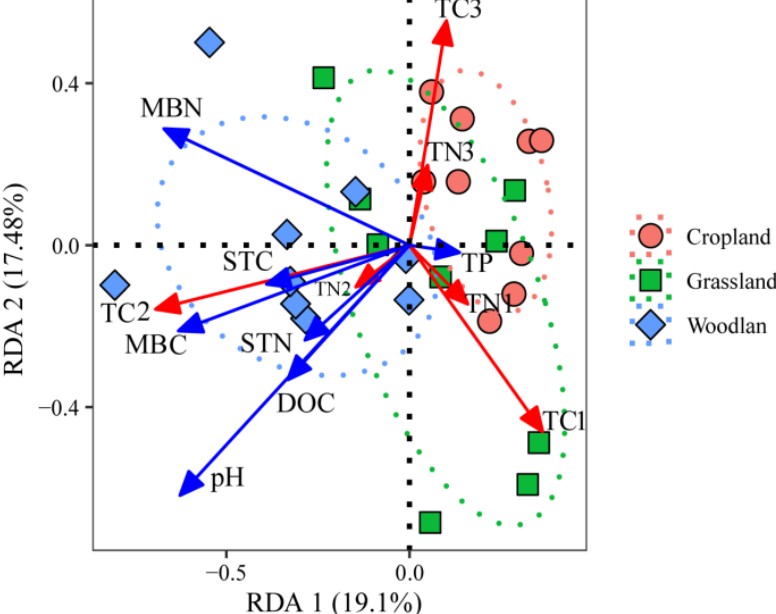

**Figure 5.** Ordination plot of RDA for the soil particle-size-associated carbon and nitrogen with soil physicochemical properties as the explaining variables. The solid points indicate abandoned cropland

scores; solid squares indicate grassland scores; solid rhombuses indicate woodland scores. The ellipses indicate the standard errors ($n = 9$) of types of land use with a 95% confidence level. STC is soil total carbon; STN is soil total nitrogen; TP is soil total phosphorus; DOC is dissolved organic carbon; MBC is microbial biomass carbon; and MBN is microbial biomass nitrogen. TC1 and TN1 are sand particle-size-associated carbon and nitrogen content, respectively; TC2 and TN2 are silt particle-size-associated carbon and nitrogen content, respectively; and TC3 and TN3 are clay particle-size associated carbon and nitrogen content, respectively.

We found that MWD and particle size fractions were closely related to the corresponding soil physicochemical properties and particle-size-associated nutrient (Table 3). Specifically, DOC positively correlated with MWD, while MBC negatively correlated with MWD. In addition, MBC positively correlated with silt particle but negatively correlated with sand particle. In the multiple regression equation of sand particle, silt particle, and MWD, TC2 was the main variable that influenced the equation (Table 3).

**Table 3.** Multiple linear regression analysis of particle-size fractions, MWD, particle-size-associated nutrients, and soil physicochemical properties of different historical land-use types in newly formed reservoir buffer strips after water impoundment.

| Aggregate Properties | Linear Regression Model with Partial Correlation Coefficient | $R^2$ |
|---|---|---|
| Sand | $21.120 - 15.751TN_1 (21.15) + 5.102TC_2 (49.27) - 3.384TC_3 (16.04) + 21.925TN_3 (7.76) + 44.428STP (5.98) + 1.135DOC (1.87) - 0.178MBC (10.90)$ | 0.8618 |
| Silt | $37.597 + 13.974TN_1 (10.84) - 4.359TC_2 (25.26) + 0.903TC_3 (2.87) - 25.209STP (1.91) + 0.169MBC (6.51)$ | 0.7044 |
| Clay | $45.08 - 9.665TN_2 (10.79) + 0.873TC_3 (2.78) - 13.028TN_3 (5.83) - 0.482STC (4.68) - 15.571STP (1.86)$ | 0.4556 |
| MWD | $230.063 - 157.679TN_1 (21.25) + 51.162TC_2 (49.70) - 34.269TC_3 (16.49) + 222.819TN_3 (8.04) + 447.551STP (6.09) + 11.472DOC (1.91) - 1.776MBC (10.90)$ | 0.8632 |

All variables are partially correlated with MWD, sand particle, silt particle, clay particle; the partial regression sum of squares is shown in parentheses. TC1 and TN1 are sand particle-size-associated carbon and nitrogen content, respectively; TC2 and TN2 are silt particle-size-associated carbon and nitrogen content, respectively; and TC3 and TN3 are clay particle-size-associated carbon and nitrogen content, respectively. STP is soil total phosphorus; DOC is dissolved organic carbon; and MBC is microbial biomass carbon.

A more in-depth analysis using PLS-PM and showing the direct and indirect effect of historical land-use types and distance from the bank on sand, silt, and clay particle-size associated nutrients, microbial activity, and soil-chemical properties is needed (Figure 6). Distance from the water on sand particle-size-associated nutrients had the highest path coefficients (0.78, $p < 0.0001$). The effect of silt particle-size-associated nutrient and clay particle-size-associated nutrients on microbial activity had relatively high path coefficients ($-0.59$, $p = 0.019$; 0.69, $p = 0.036$). The goodness of fit (GOF) was 0.49, which indicated that the model had relatively good predictive power.

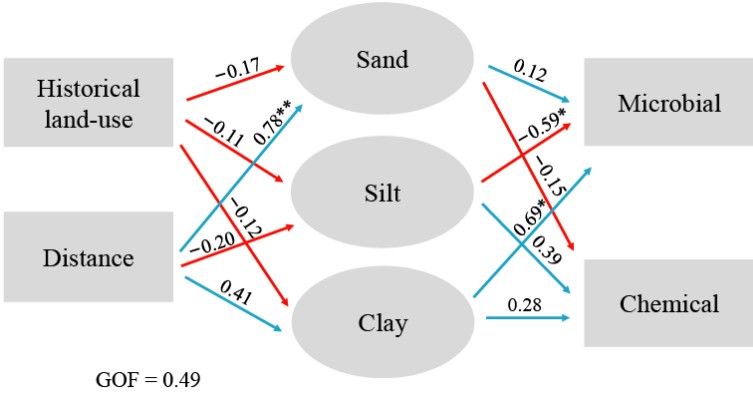

**Figure 6.** The relationship between historical land-use types, distance from the water, sand particle-size-associated nutrient content, silt particle-size-associated nutrient content, clay particle-size associated

nutrient content, soil microbial activity, and soil chemical properties based on PLS-PM. Historical land-use types (constructed using a dummy variable); distance, distance from the water; sand, sand particle-size-associated nutrients (constructed using sand particle-size-associated carbon, sand particle-size-associated nitrogen, and sand-particle content); silt, silt particle-size-associated nutrients (constructed using silt particle-size-associated carbon, silt particle-size-associated nitrogen, and silt-particle content); clay, clay particle-size-associated nutrients (constructed using clay particle-size-associated carbon, clay particle-size-associated nitrogen, and clay-particle content); microbial, microbial activity (constructed using MBC, MBN, and DOC); chemical, soil chemical properties (constructed using STC, STN, STP, and pH). Blue and red arrows represent positive and negative pathways, respectively. The standard path coefficients are shown on the arrow. * represents significance at 0.05 level. ** represents significance at 0.01 level.

## 4. Discussion

The establishment of reservoir buffer strips is a management measure that has been widely promoted worldwide as an effective way to reduce pollution and improve water quality [15]. In this study, different historical land-use types in the buffer strips of a newly built reservoir experienced complex dynamics in soil aggregate stability, particle-size-associated nutrient distribution characteristics, and geochemical processes after water impoundment, showing different response mechanisms at different distance from the water.

### 4.1. Soil Aggregates Stability Characteristics of Historical Land-Use Types in Reservoir Buffer Strips

In this study, we classified the soil texture of the woodland as sandy loam and the abandoned cropland and grassland as silt loam. We found that the MWD of woodland was significantly greater than that of the other two land-use types, indicating that woodland had greater soil-aggregate stability and was better able to resist the effects of hydraulic erosion [53,54].

Many studies have reported the response mechanisms of soil particle-size fractions and soil-aggregate stability in the reservoir riparian zone during seasonal dynamics and long-term alternating dry-wet cycles [8,55]. Therefore, the process of soil-aggregate stability in response to inundation is still controversial. Sarah and Rodeh (2004) [56] suggested that soil would gradually transition toward a typical nature "steady state" during wetting, and soil-aggregate stability would increase. The more common view is that after repeated dry-wet cycles, the shrinkage and expansion of the soil due to hydrological stress leads to a stabilization of the particle-size distribution and a decrease of aggregate stability [30]. E.g., Nsabimana et al. [8] studied the impacts of water level fluctuations on soil-aggregate stability in the Three Gorges Reservoir and found that MWD decreased gradually along the elevation. The reasons for these two differences may be closely related to the study region and soil submerged conditions. In this study, both grassland and woodland showed a tendency for MWD at 0 m to be greater than at 2 m and 20 m. The soil at 0 m had been submerged since water impoundment and was rarely exposed, which suggests that short-term wetting facilitates soil aggregate stability, and this assumption is consistent with the findings of Denef et al., (2001) [39], Totsche et al., (2017) [14], and others. The MWD at 0 m of abandoned cropland is greater than 2 m and less than 20 m, which may be related to less aboveground vegetation coverage in this area [57,58].

MWD at 2 m is lower than at 0 m and 20 m for the different historical land-use types, suggesting that water impoundment reduces soil-aggregate stability in the waterward location of the reservoir buffer strips [8,59]. This finding may be mainly due to the fact that 2 m is closer to the bank where the soil is frequently in alternating dry-wet cycles caused by groundwater table elevation and surface runoff, which makes the soil at this location more susceptible to disintegration and thus reduces soil-aggregate stability [60,61].

In summary, the newly formed reservoir buffer strips of different historical land-use types showed significant differences in soil-aggregate stability at different distances after

reservoir impoundment because of the dry-wet cycles and water submerge condition caused by different distance from water and, which is consistent with our hypothesis.

*4.2. Particle-Size Associated Nutrient Distribution of Historical Land-Use Types in Reservoir Buffer Strips*

Soil organic matter (SOM) is an essential cementing substance in soil aggregates and determines the formation of water-stable aggregates, and the size and mass percentage of soil particles significantly affect the size and quantity of soil aggregates and their binding to organic matter [62,63]. Soil organic carbon obtained through soil particle-size grouping is an effective entry point to understand further the relationship between the decomposition transformation and stability of the SOM and the location and state in which it exists [64,65].

In this study, without consider 0 m site, we observed that the carbon content in sand particle was significantly lower than that in silt particle of both grassland and woodland; nevertheless, soil particle-size associated carbon distribution at 0 m may be influenced by organic matter carried by overlying water [66]. Although the carbon content in sand particle at W0 and W2 of the abandoned cropland was greater than that of the clay particle, the difference was not statistically significant probably because the carbon content of the soil macro-aggregate increased during previous cultivation [33,67]. In general, in the restored soil types, the organic carbon and nitrogen that can be bound by the soil macro-aggregate was greater than that of the micro-aggregate [68,69]. We speculated that the loss of carbon in sand particle of historical land-use types in reservoir buffer strips occurred to different degrees after water impoundment possibly because the sand particle was the main source of the soil active carbon pool and, therefore, more vulnerable to the risk of loss by external erosion such as water inundation [7,70].

The response of soil particle-size-associated carbon to external erosion, such as inundation, is accompanied by fragmentation and regeneration of different particle-size fractions, and the direction of evolution depends on which process is dominant [31,39,71]. Soil organic matter adsorption on clay particle is an important determinant of soil organic matter stability, and fine silt-clay particles (<20 $\mu$m) have the maximum capacity to bind soil organic carbon in different soil types. Therefore, one of the main factors in the physical conservation of soil organic matter is its ability to bind with fine silt-clay fractions [37,72]. Among them, the silt particle are the intermediaries of soil organic carbon turnover, and when disturbed by external forces, the microorganisms attached to the silt particle can use the organic carbon isolated by the silt particle to accelerate the nutrient turnover between different particle sizes [14,32,70,73]. In this study, the silt particle-size-associated carbon content was generally low except in W20, while the silt-particle content in woodland was the lowest, which indicated that soil silt particle in the woodland had a high carbon-binding capacity. In addition, the soil silt particle-size-associated carbon content of woodland showed greater than sand particle-size-associated carbon content, and abandoned cropland and grassland showed the opposite trend. This possibly because the woodland had a more abundant root system, which enhanced soil aeration, and there could be two different response mechanisms at different distance from the water in woodland because of the effect of the water table. In the waterward region, the soil condition was affected by water dam-triggered flood intensity, nutrient turnover between soil particles was accelerated, and the silt particle-size-associated carbon content was reduced. While in the landward region, which was less stressed by water inundation, clay particles were still able to provide good physical protection for the soil organic matter. In contrast, both abandoned cropland and grassland were affected by water storage to varying degrees, and nutrient turnover between different particle sizes was accelerated. Extensive studies have shown that the width of reservoir buffer strips should vary according to different land-use types [74,75]. Therefore, we speculated that the 20 m we set for abandoned cropland and grassland may not completely cover the dam-triggered flooding impact area, and the width setting for the abandoned cropland and grassland buffer strips should be expanded for practical restoration.

In this study, we also observed that soil total carbon (STC) was significantly higher in W2 than in W20 probably because, after reservoir impoundment, ample organic matter in the woodland migrated from upper to lower areas with surface runoff, and the fluctuation in water level also brought a large amount of organic matter to the bank, creating a hot spot for carbon accumulation in the waterward region [8,76]. However, the particle-sizes associated carbon content was significantly higher in W20 than in W2, and there was apparently a large margin in between, which was not significantly observed in other historical land-use types. A possible explanation is that, after reservoir impoundment, the release of organic matter exists in the form of inorganic carbon, and the carbonate forms precipitate with some metal ions in the water, which are stored on the surface of the sediment [77–79]. This situation also explains the low OCP we calculated in W2, which suggests that the use of particle sizes grading to estimate carbon stocks in newly formed reservoir buffer strips may result in an underestimation of woodland carbon sequestration.

Take together, the newly formed reservoir buffer strips of different historical land-use types all underwent corresponding changes in the particle-size-associated carbon distribution characteristics after reservoir impoundment mainly due to the turnover properties of different soil particles combined with organic carbon.

### 4.3. Relationship between Soil Particle Fractions and Chemical Properties in Historical Land-Use Types in Reservoir Buffer Strips

In the process of reservoir buffer strip evolution and restoration, soil texture is affected by internal factors of the soil parent material and external factors such as soil chemical composition, climate, water, and vegetation [45,80,81]. Dam-triggered flooding intensity also determines the trend of soil texture and nutrients [51,63]. The dynamics of nutrients (e.g., nitrogen, phosphorus, and potassium) can indirectly affect the stability of the soil aggregates and the associated carbon-distribution patterns [82].

Reviewing the relationship between soil aggregate stability and SOM, Six and Paustian (2014) [83] stated that, for a long time, SOM physical fractionation emphasized the influence of soil substrate on SOM pool size and dynamics, but did not have a good correlation with soil-chemical properties. In this study, we found that sand particle, silt particle, and MWD were well correlated with soil physicochemical properties and particle-size-associated carbon content ($R^2 = 0.86$, $R^2 = 0.74$, $R^2 = 086$), and $TC_2$ was the most significant determinant. The results of RDA and PLS-PM analyses showed that the distribution of silt particle-size-associated carbon was closely related to soil microbial activity, which suggested that soil microbial activity was an important factor influencing the distribution of soil particle-size-associated carbon content in newly formed reservoir buffer strips. It is well known that soil microorganisms play an essential role in the formation, stabilization, and destruction of soil aggregates [84]. Different microbial communities prefer to colonize different soil particle-size fractions and are adapted to the physical and chemical properties of the soil [73,74,85]. This difference in preference may be due to the transition from aerobic to anaerobic soil microbial metabolism, which evolved to adapt in specific environments for different historical land-use types in the reservoir buffer strips after reservoir impoundment where microbial activity becomes a major limiting factor for soil ecology function [77]. In addition to underscoring the important role of nutrient turnover in silt fraction, we can assert that reservoir impoundment accelerates the turnover of silt particle-size-associated nutrient in soils of historical land-use types in the reservoir buffer strips, and this process may be mediated mainly by microbial activity.

It is worth noting that the sampling time in this study was carried out two months after the reservoir reached its baseline (normal level), only one sampling was carried out. The limitation of this study is that it can only represent the soil characteristic after short-term inundation. The long-term evolution mechanism of soil particle-size-associated carbon cannot be deeply explored. Second, due to the patchy distribution characteristics of historical land-use types around the reservoir and the time limit for conducting experiment,

it is difficult to obtain the soil background value before water impoundment, and there is lack of effective comparison.

## 5. Conclusions

The stability of soil aggregates and organic matter distribution characteristics are key mechanisms in understanding the evolution of the environment in and around reservoirs; a better ecological understanding can aid in the management of reservoir buffer strips in general [86]. In this study, we found that soil aggregate stability and particle-size-associated carbon distribution characteristics of different historical land-use types in newly formed reservoir buffer strips showed different response mechanisms after reservoir impoundment, which were mainly caused by the dry-wet cycle and water submerge condition at different distance from water- and soil-associated carbon-turnover mechanisms of different particle-size fractions.

The impact of dam construction on reservoir environments is long-lasting and far-reaching [87]; different historical land-use types around a reservoir gradually transition to an aquatic ecosystem, so this study is an important guide to design and restore reservoir buffer strips. In future reservoir buffer-strip management, different protection measures and width settings should be adopted for different historical land-use types, and high priority should be given to the protection of waterward regions. In particular, the reservoir buffer strips of Chushandian Reservoir have become a popular destination for fishing enthusiasts, and excessive human interference has aggravated the loss and degradation of soil nutrition, which greatly threatens the ecological safety and water quality of the reservoir bank. We focused on soil aggregates stability and associated organic matter distribution characteristics of different historical land-use types in reservoir buffer strips after reservoir impoundment, but the long-term mechanism of soil ecology with seasonal drainage and storage warrants further study.

**Author Contributions:** T.Y. and B.Z. designed the experiment, participated in the survey, analysed the data, and wrote the manuscript. Y.K. and S.H. supervised the project and wrote the manuscript. X.W., Z.Y., Q.H., M.F., X.L. and Z.W. participated in the fieldwork. All authors discussed the results, provided critical feedback, and helped shape the research. All authors have read and agreed to the published version of the manuscript.

**Funding:** This research was funded by the Xinyang Agricultural and Forestry University Science and Technology Innovation Team project (CXTD-201904).

**Institutional Review Board Statement:** Not applicable.

**Informed Consent Statement:** Not applicable.

**Data Availability Statement:** Data are available from the corresponding author and will be provided upon reasonable request.

**Acknowledgments:** We appreciate the assistance provided by the Chushandian Reservoir Management Bureau.

**Conflicts of Interest:** There is no conflict of interest/competing interest.

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
