# Peer review of "The Relationship between Soil Particle Size Fractions, Associated Carbon Distribution and Physicochemical Properties of Historical Land-Use Types in Newly Formed Reservoir Buffer Strips"

_sustainability, doi:10.3390/su14148448_

Round 1
Reviewer 1 Report
The authors studied the soil particle fractions, particles stability, and particle-size associated carbon distribution characteristics of different land use types, such as abandoned cropland, grassland and woodland from the reservoir buffer strips at distance scale before the construction and impoundment of the Chushandian Reservoir, China. The main purpose was to explore the relationship between soil particles mass proportion and associated carbon and physicochemical properties. The manuscript is very interesting. However, I have some concerns, which need to be addressed before considering for final publication.
Point 1: Title is not appreciated. The current title was obscure.
Point 2: The primitive characteristics of the soil in abandoned cropland, grassland and woodland before the construction of the reservoir are needed.
Point 3: The Introduction Section was too long, which need to be succinct and clear.
Point 4: The Figure 1 was indistinct, and all the tables are not in accordance with the requirements of Sustainability.
Point 5: References are too old.
Point 6: Line 521, what does TC2 mean?
Point 7: In Discussion section, you should discuss limitations of this study.
Author Response
Response to Reviewer 1 Comments:
Comments and Suggestions for Authors
The authors studied the soil particle fractions, particles stability, and particle-size associated carbon distribution characteristics of different land use types, such as abandoned cropland, grassland and woodland from the reservoir buffer strips at distance scale before the construction and impoundment of the Chushandian Reservoir, China. The main purpose was to explore the relationship between soil particles mass proportion and associated carbon and physicochemical properties. The manuscript is very interesting. However, I have some concerns, which need to be addressed before considering for final publication.
Response: Thanks for you affirmation of this study, and thanks for your valuable comments on this manuscript, which play a very import role in improving the quality of our manuscript.
Point 1: Title is not appreciated. The current title was obscure.
Response 1: we have changed the title to “The relationship between soil particle size fractions, associated carbon distribution and physicochemical properties of historical land-use types in newly formed reservoir buffer strips”, which is more in line with our research. As for the previous title, we want to make our manuscript seems more attractive.
Point 2: The primitive characteristics of the soil in abandoned cropland, grassland and woodland before the construction of the reservoir are needed.
Response 2: we added the data of primitive characteristic of the soil before the reservoir was constructed as “The primitive characteristics of the soil in three land-use types before reservoir construction was pH: 7.1; TC (total carbon): 8.32g/Kg; EC (electric conductivity): 55.40 μS·cm-1, respectively [44].”, which the data from literature.
- Xie D. B.; Wang, P.; Wang, T. F.; Niu, H. D.; Chu, J.; Lei, Y. K. Plant configuration based on spatial analysis of soil parameters in the dam site of Chushandian Reservoir, Northern Horticulture, 2020, 11, 91-98. (Abstract in English)
We are very sorry that due to the time of sampling and patchy characteristic of the historical land-use types in newly formed reservoir buffer strips, we could not sample the background value of the soil before the reservoir was impoundment. We have supplemented this limitation in the discussion section.
Point 3: The Introduction Section was too long, which need to be succinct and clear.
Response 3: we partially deleted and rearranged the Introduction Section, and adjusted the sequence of paragraphs to make the connection between the chapters closer, we also emphasized the ideology of experiment and what gap it cover.
Point 4: The Figure 1 was indistinct, and all the tables are not in accordance with the requirements of Sustainability.
Response 4: we reedited Figure 1, bolded the latitude and longitude, and saved it to a high-assurance Tif format. At the same time, the table of the manuscript was modified according to the requirements of Sustainability.
Point 5: References are too old.
Response 5: we replaced all the literature before 2000 with the most recent literature, except for two related to the theory originally proposed.
Point 6: Line 521, what does TC2 mean?
Response 6:TC2 is the silt particle-size associated carbon content. Thanks for you kindly reminding. We marked all abbreviation the first time they appeared.
Point 7: In Discussion section, you should discuss limitations of this study.
Response 7: After careful consideration, we wrote the limitation of this study as follows in Discussion section: “It is worth noting that the sampling time in this study was carried out two months after the reservoir reached its baseline (normal level), only one sampling was carried out. The limitation of this study is that it can only represent the soil characteristic after short-term inundation. The long-term evolution mechanism of soil particle-size associated carbon cannot be deeply explored. Secondly, due to the patchy distribution characteristics of historical land-use types around the reservoir and the time limit for conducting experiment, it is difficult to obtain soil background value before water impoundment, and there is lack of effective comparison”. We hope this will meet requirements.
Reviewer 2 Report
The reservoir buffer strips were indeed important in maintaining the local ecosystem and environment. The manuscript aims to explore the relationship between soil particles mass proportion and its associated carbon and physicochemical properties by using different characters of soil particle-size. The research is meaningful and the result is significant. To my knowledge, the grain-size endmember model built by Zhang et al. 2018 (see below) would be a good method when it related to the grain-size characters of sediment.
Zhang Xiaonan, Zhou Aifeng, Wang Xin, Song Mu, Zhao Yongtao, Xie Haichao, Russell James M., Chen Fahu, 2018. Unmixing grain size distributions in lake sediments: a new method of endmember modeling using hierarchical clustering. Quaternary Research, 89(1): 365-373.
Author Response
Response to Reviewer 2 Comments
The reservoir buffer strips were indeed important in maintaining the local ecosystem and environment. The manuscript aims to explore the relationship between soil particles mass proportion and its associated carbon and physicochemical properties by using different characters of soil particle-size. The research is meaningful and the result is significant. To my knowledge, the grain-size endmember model built by Zhang et al. 2018 (see below) would be a good method when it related to the grain-size characters of sediment.
Zhang Xiaonan, Zhou Aifeng, Wang Xin, Song Mu, Zhao Yongtao, Xie Haichao, Russell James M., Chen Fahu, 2018. Unmixing grain size distributions in lake sediments: a new method of endmember modeling using hierarchical clustering. Quaternary Research, 89(1): 365-373.
Response: Thanks for your affirmation of this manuscript. We have seriously read the literature you recommended, it is very helpful to improve the quality of our manuscript, and we have also cited it in the manuscript.
Reviewer 3 Report
Too long abstrct and Introduction.
What do you mean under particles stability? And what under aggregate stabilty? Using as synonyms are not acceptable!
Please, apply commonly-used phrases.
What do you mean under „soil particle mass proportion”? If soil particle size distribution, please use that! In Google I could fine 11 results, only!
There is no clear hypothesis.
Results are not well-explained in every cases.
However they cited good and right references in right places, they don’t use the correct phrases.
Did you measure the pH of the different soil fractions, and make correlation with it?
According to USDA texture triangle, I got different texture classes as the author said.
For silt fraction they mention these thresholds:
- 2-3 microm (L. 233.)
-2-53 microm (in Table 2.)
-2-20 microm (L. 465.)
Which is true?
Author Response
Response to Reviewer 3 Comments
Point1: Too long abstract and Introduction.
Response 1: We have appropriately deleted and organized the Abstract and Introduce sections, rearranged the order of the paragraphs, and made the connection between the preceding and following chapters more closely. At the same time, we have also emphasize what the gap it cover and importance of the current research field of this manuscript.
Point 2: What do you mean under particles stability? And what under aggregate stability? Using as synonyms are not acceptable! Please, apply commonly-used phrases.
Response 2: Thanks a lot for your suggestion, we changed all “particles stability” to “aggregates stability”.
Point 3: What do you mean under soil particle mass proportion”? If soil particle size distribution, please use that! In Google I could fine 11 results, only!
Response 3: We apologize for the confusion. Our intention is to use “soil particle mass proportion” express the “soil particle size content”. Maybe this is not a common terminology. We changed all the “soil particle mass proportion” to “particle size fractions”, which is a more common used terminology. We were going to use the terminology you recommended, but considered the subsequent expression “particle-size associated carbon distribution”, in order to make the language seem more fluent, we used the former.
Point 4: There is no clear hypothesis.
Response 4: After careful consideration, we reorganized the hypothesis as: “We hypothesize that the response to of soil aggregates stability to water impoundment differs on the distance scales in the newly formed reservoir buffer strips, and that this difference is mainly caused by dry-wet cycle at different distance from water; In addition, we assume that water impoundment will change the binding characteristic of soil particles with organic carbon in newly formed reservoir buffer strips, and there is a close relationship with soil physicochemical properties.”
Point 5: Results are not well-explained in every cases.
Response 5: Thanks for your kindly reminding. After carful comparison, there is really some results not well-explained in the Discussion section. Especially in term of soil aggregates stability, we re-discussed the manuscript in the light of experimental Results. In addition, we proofread the Results and Discussion sections carefully, we found that the distribution characteristic of particle –size associated carbon at 0m site seems don’t show as much regularity as the other site, we think this maybe due to submerge condition at this location, soil particle-size associated carbon distribution at 0m may influenced by organic matter carried by overlying water. At the same time, we have made modification to some other related issues not well-explained we found.
Point 6: However they cited good and right references in right places, they don’t use the correct phrases.
Response 6: The format and location of references have been carefully revised and checked for the requirements of Sustainability.
Point 7: Did you measure the pH of the different soil fractions, and make correlation with it?
Response 7: I am sorry, we don’t measure the pH of the different soil fractions. Because we used wet sieving method to separate the soil particles, it is difficult to measure the pH of different soil fractions. The pH we measured in this manuscript is whole soil.
Point 8: According to USDA texture triangle, I got different texture classes as the author said.
Response 8: After carefully comparing our results with the USDA soil texture triangle, we really get the soil texture classification of abandoned cropland and grassland wrong. We have changed the soil texture of abandoned cropland and grassland as silt loam in the manuscript. Here I also express my admiration for the rigorous scientific attitude of the reviewer.
Point 9: For silt fraction they mention these thresholds:
- 2-3 microm (L. 233.)
-2-53 microm (in Table 2.)
-2-20 microm (L. 465.)
Which is true?
Response 9: I am sorry we mistake the units of soli particle size, the threshold of silt fraction should be 2-53μm. In the L.465, the < 20μm which we want to express is the fine silt-clay particles. We have thoroughly checked and revised similar errors.
Reviewer 4 Report
Dear Sir,
Greetings!
Thank you so much for your invitation to review the manuscript “Soil microbial activity mediated the turnover of silt particle-size carbon of historical land-use types in newly formed reservoir buffer strips”. The present paper is much suitable to the journal. However there is still need some improvement that are as follow.
## The Abestract need to be concise perticularly the section. We focused on soil particle fractions,particles stability, and particle-size associated carbon distribution characteristics of different his-
torical land-use types of reservoir buffer strips at distance scale (i.e., different distance from the water) after reservoir impoundment in the Chushandian Reservoir, China, and explored the relationship between soil particles mass proportion and associated carbon and physicochemical properties. The results showed that the soil texture of abandoned cropland and grassland are classified as loam and woodland are classified as sandy loam; MWD at 2 m was lower than at 0 m and 20 m for different historical land-use types, which indicated that reservoir impoundment reduced soil particles stability in the waterward region of the reservoir buffer strips.
## The introdcution is well but need to improve with the ideology of the experiment and what gap it cover.
## Latitude and Longitude of Fig 1 is not clear. Figure 1. Location of study area. (top left) Boundary map of China. (top right) Boundary map of Henan province. (bottom left) Locations of the sampling sites in the riparian zone of the Chushan dian Reservoir, China. (bottom right) Sample strip setting diagram. The pentagram represents the abandoned cropland; the diamond represents the grassland; and the triangle represents the woodland. The circle represents the setting of sampling strips.
Other things are well and good.
Thanks much
Author Response
Response to Reviewer 4 Comments
Thank you so much for your invitation to review the manuscript “Soil microbial activity mediated the turnover of silt particle-size carbon of historical land-use types in newly formed reservoir buffer strips”. The present paper is much suitable to the journal. However there is still need some improvement that are as follow.
Response: Thanks for you affirmation of this study, and thanks for your valuable comments on this manuscript, which play a very import role in improving the quality of our manuscript.
Point 1: The Abstract need to be concise particularly the section. We focused on soil particle fractions, particles stability, and particle-size associated carbon distribution characteristics of different historical land-use types of reservoir buffer strips at distance scale (i.e., different distance from the water) after reservoir impoundment in the Chushandian Reservoir, China, and explored the relationship between soil particles mass proportion and associated carbon and physicochemical properties. The results showed that the soil texture of abandoned cropland and grassland are classified as loam and woodland are classified as sandy loam; MWD at 2 m was lower than at 0 m and 20 m for different historical land-use types, which indicated that reservoir impoundment reduced soil particles stability in the waterward region of the reservoir buffer strips.
Response 1: We have simplified the Abstract section, highlighted the main research results, and shortened some sentences to make the structure between the front and back more compact.
Point 2: ## The introduction is well but need to improve with the ideology of the experiment and what gap it cover.
Response 2: We have appropriately deleted and organized the Introduce sections, rearranged the order of the paragraphs, and made the connection between the preceding and following chapters more closely. At the same time, we have also emphasize what the gap it cover and importance of the current research field of this manuscript.
Point3: ## Latitude and Longitude of Fig 1 is not clear. Figure 1. Location of study area. (top left) Boundary map of China. (top right) Boundary map of Henan province. (bottom left) Locations of the sampling sites in the riparian zone of the Chushandian Reservoir, China. (bottom right) Sample strip setting diagram. The pentagram represents the abandoned cropland; the diamond represents the grassland; and the triangle represents the woodland. The circle represents the setting of sampling strips.
Response 3: we reedited Figure 1, bolded the latitude and longitude, and saved it to a high-assurance Tif format.